

# Bombardiers and assassins: mimetic interactions between unequally defended insects

Shinji Sugiura[1] and Masakazu Hayashi[2]

[1] Graduate School of Agricultural Science, Kobe University, Kobe, Hyogo, Japan
[2] Hoshizaki Green Foundation, Izumo, Shimane, Japan

## ABSTRACT

In defensive mimicry, resemblance between unequally defended species can be parasitic; this phenomenon has been termed quasi-Batesian mimicry. Few studies have used real co-mimics and their predators to test whether the mimetic interactions were parasitic. Here, we investigated the mimetic interaction between two well-defended insect species, the bombardier beetle *Pheropsophus occipitalis jessoensis* (Coleoptera: Carabidae) and the assassin bug *Sirthenea flavipes* (Hemiptera: Reduviidae), using their potential predator, the pond frog *Pelophylax nigromaculatus* (Anura: Ranidae), which coexists with these insect species in the same habitat in Japan. We observed behavioural responses of this frog species (adults and juveniles) to adult *Ph. occipitalis jessoensis* and adult *S. flavipes* under laboratory conditions. Among the frogs, 100% and 75% rejected *Ph. occipitalis jessoensis* and *S. flavipes*, respectively, suggesting that, compared with the assassin bug *S. flavipes*, the bombardier beetle *Ph. occipitalis jessoensis* is more well-defended against frogs. An assassin bug or a bombardier beetle was provided to a frog that had encountered the other insect species. Frogs with a history of assassin bug encounter demonstrated a lower rate of attack toward bombardier beetles. Similarly, frogs with a history of bombardier beetle encounter demonstrated a lower rate of attack toward assassin bugs. Therefore, both the bombardier beetle *Ph. occipitalis jessoensis* and the assassin bug *S. flavipes* benefit from the mimetic interaction.

Corresponding author
Shinji Sugiura,
sugiura.shinji@gmail.com,
ssugiura@people.kobe-u.ac.jp

## INTRODUCTION

Animals have evolved diverse anti-predator strategies such as chemical, morphological, physical, and behavioural defences (*Edmunds, 1974*; *Ruxton, Sherratt & Speed, 2004*; *Eisner, Eisner & Siegler, 2005*; *Sugiura, 2020a*). Some well-defended animals have aposematic body colours, which signal distaste and danger to their predators (*i.e.,* warning signals; *Ruxton, Sherratt & Speed, 2004*). Well-defended species frequently share warning signals that serve to deter predation (*i.e.,* Müllerian mimicry; *Müller, 1878*; *Müller, 1879*; *Ruxton, Sherratt & Speed, 2004*; *Sherratt, 2008*), while some non-defended species mimic well-defended species (*i.e.,* Batesian mimicry; *Bates, 1862*; *Ruxton, Sherratt & Speed, 2004*). Müllerian mimicry is the mutualistic interaction between equally defended species, while Batesian mimicry

is the parasitic or commensal interaction between non-defended and well-defended species (*Ruxton, Sherratt & Speed, 2004*; *Sherratt, 2008*; *Balogh, Gamberale-Stille & Leimar, 2008*; *Honma, Takakura & Nishida, 2008*). These two types of mimicry are traditionally considered extreme ends of a defensive mimicry spectrum (*Balogh, Gamberale-Stille & Leimar, 2008*). On the mimicry spectrum, mimetic interactions between unequally defended species can be parasitic—this phenomenon is termed quasi-Batesian mimicry (*Speed, 1993*; *Speed, 1999*; *Speed & Turner, 1999*; *Rowland et al., 2010*). However, there is controversy regarding whether mimetic interactions between unequally defended species are truly parasitic (*Speed et al., 2000*; *Rowland et al., 2007*; *Rowland et al., 2010*; *Aubier, Joron & Sherratt, 2017*). Some experimental studies have indicated such interactions are parasitic (*Speed et al., 2000*; *Rowland et al., 2010*), while other studies have indicated that they are mutualistic (*Rowland et al., 2007*) or not always parasitic (*Lindström et al., 2006*). Previous experimental studies used bird predators and artificial prey to investigate the nature of quasi-Batesian mimicry (*Speed et al., 2000*; *Lindström et al., 2006*; *Ihalainen, Lindström & Mappes, 2007*; *Rowland et al., 2007*; *Ihalainen et al., 2008*; *Rowland et al., 2010*). Although resemblance between unequally defended species is commonly found in the natural environment (*Marples, Brakefield & Cowie, 1989*; *Marples, 1993*; *Winters et al., 2018*; *Chouteau et al., 2019*; *Soukupová, Veselý & Fuchs, 2021*), few studies have used real co-mimics and their natural predators to determine whether the mimetic interactions are parasitic or mutualistic (*Pekár et al., 2017*; *Raška et al., 2020*).

Bombardier beetles (Coleoptera: Carabidae: Brachininae: Brachinini) are chemically defended; their adults eject toxic chemicals at a temperature of 100 °C when they are attacked by predators (*Aneshansley et al., 1969*; *Dean, 1979*; *Eisner, Eisner & Siegler, 2005*; *Arndt et al., 2015*). The discharge of hot chemicals—namely, bombing—can protect beetles from various groups of predators such as birds (*Kojima & Yamamoto, 2020*), reptiles (*Bonacci et al., 2008*), amphibians (*Eisner & Meinwald, 1966*; *Dean, 1980*; *Sugiura & Sato, 2018*; *Sugiura, 2018*; *Sugiura & Date, 2022*), and arthropods (*Eisner, 1958*; *Eisner & Meinwald, 1966*; *Eisner & Dean, 1976*; *Eisner et al., 2006*; *Sugiura, 2021*). Many bombardier beetle species have similar aposematic body colours (*Schaller et al., 2018*; *Anichtchenko et al., 2022*) and are visually mimicked by some insect species that coexist with them in the same habitats (*Shelford, 1902*; *Bonacci et al., 2008*; *Kojima & Yamamoto, 2020*). For example, in Italy, the carabid species *Anchomenus dorsalis* (Pontoppidan) and the bombardier beetle species *Brachinus sclopeta* (Fabricius) have a similar body colour pattern (green-blue and red-brown) (*Bonacci et al., 2008*; *Bonacci, Brandmayr & Brandmayr, 2011*). In Borneo, a raspy cricket species (Orthoptera: Gryllacrididae) shares a black and orange body colour with the bombardier beetle *Pheropsophus* (*Stenaptinus*) *agnatus* (Chaudoir) (*Shelford, 1902*). In Japan, the assassin bug *Sirthenea flavipes* (Stål) (Hemiptera: Reduviidae: Peiratinae) has a black and yellow body colour similar to the colour of the bombardier beetle *Pheropsophus* (*Stenaptinus*) *occipitalis jessoensis* Morawitz (formerly called *Pheropsophus jessoensis*; Figs. 1A and 1B; *Kojima & Yamamoto, 2020*). Although the interaction between the carabid species *A. dorsalis* and the bombardier beetle *B. sclopeta* has been suggested to constitute Müllerian mimicry (*Bonacci et al., 2008*; *Bonacci, Brandmayr*

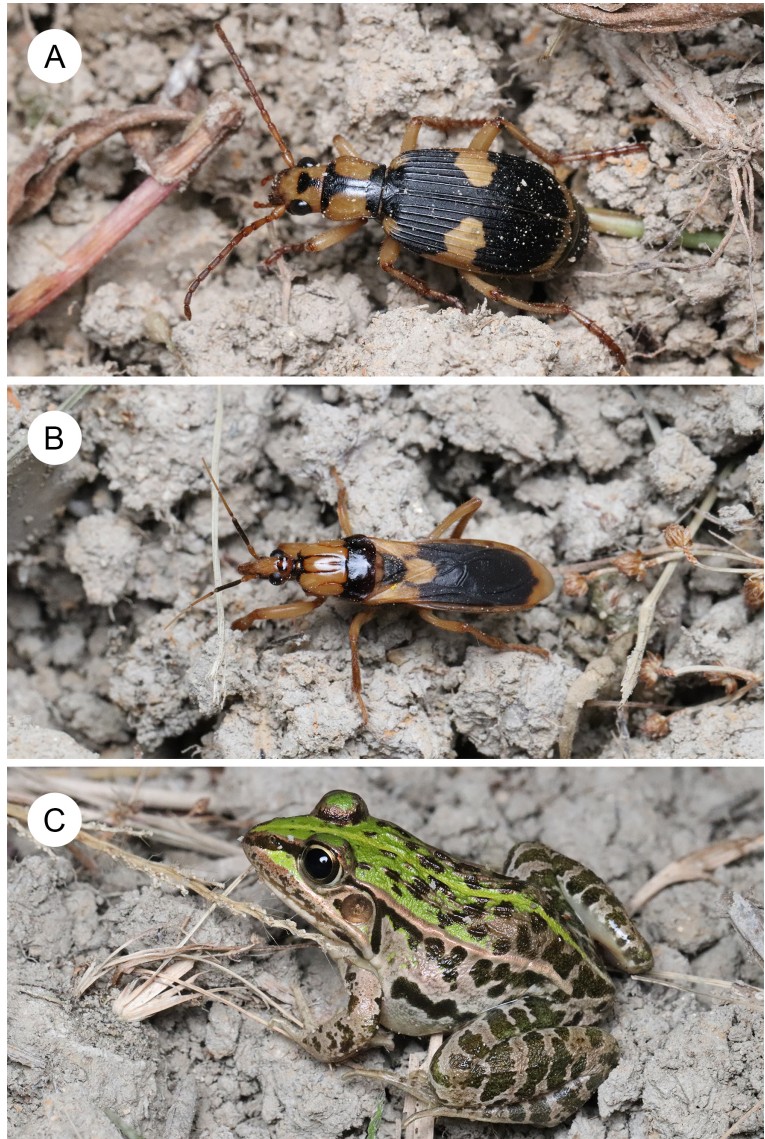

**Figure 1** **A bombardier beetle, an assassin bug, and their potential predator.** (A) An adult bombardier beetle *Pheropsophus occipitalis jessoensis*. (B) An adult assassin bug *Sirthenea flavipes*. (C) An adult pond frog *Pelophylax nigromaculatus*. The photos were taken in the same grassland (in Hyogo Prefecture) on 13 October 2021. Photo credit: Shinji Sugiura.

*& Brandmayr, 2011*), few studies have examined whether mimetic interactions between bombardier beetles and distantly related insects are parasitic or mutualistic.

To investigate mimetic interactions that involve bombardier beetles, we used two well-defined insect species (the bombardier beetle *Ph. occipitalis jessoensis* and the assassin bug *S. flavipes*) and their potential predator, the black-spotted pond frog *Pelophylax nigromaculatus* (Hallowell) (Anura: Ranidae), which coexists with these insects in the same habitat in Japan (Fig. 1).

The bombardier beetle *Ph. occipitalis jessoensis* (Fig. 1A) is distributed in East Asia (Japan, Korea, and China) and Southeast Asia (Vietnam) (*Fedorenko, 2021*). In Japan, adult *Ph. occipitalis jessoensis* are commonly found on the ground in farmland, grassland, and forest edges (*Habu & Sadanaga, 1965*; *Yahiro et al., 1992*; *Ishitani & Yano, 1994*; *Fujisawa, Lee & Ishii, 2012*; *Ohwaki, Kaneko & Ikeda, 2015*; *Sugiura, 2018*; *Sugiura, 2021*; *Sugiura & Date, 2022*). Female adults lay eggs in soil and the hatched larvae feed exclusively on egg masses of the mole cricket *Gryllotalpa orientalis* Burmeister (Orthoptera: Gryllotalpidae) (*Akino, Sasaki & Okamoto, 1956*; *Habu & Sadanaga, 1965*). Adults feed on live and dead insects of various species on the ground (*Habu & Sadanaga, 1965*; *Sugiura, 2018*). The bombardier beetle *Ph. occipitalis jessoensis* can eject quinones (1,4-benzoquinone and 2-methyl-1,4-benzoquinone) and water (vapor) at a temperature of approximately 100 °C from the end of its abdomen (Video S1; *Kanehisa & Murase, 1977*; *Kanehisa, 1996*) to repel predators such as the pond frog *Pe. nigromaculatus* (*Sugiura, 2018*), the bullfrog *Lithobates catesbeianus* (Shaw) (Anura: Ranidae) (*Sugiura & Date, 2022*), quails (*Kojima & Yamamoto, 2020*), and praying mantises (*Sugiura, 2021*).

Similar to bombardier beetles, assassin bugs are considered well-defended insects; assassin bugs kill prey insects and defend against their predators by using their proboscis (*i.e.,* labium) to inject them with painful venoms (*Eisner, Eisner & Siegler, 2005*; *Schmidt, 2009*; *Walker et al., 2016*). In addition, assassin bugs have a variety of scent glands that can act as chemical defences (*Louis, 1974*; *Staddon, 1979*). The assassin bug *S. flavipes* (Fig. 1B) is distributed in East Asia (Japan, Korea, and China), Southeast Asia (Cambodia, Indonesia, Laos, Malesia, Myanmar, Philippines, Thailand, and Vietnam), South Asia (Bangladesh, India, Nepal, Pakistan, and Sri Lanka), and West Asia (Afghanistan and Iran) (*Chłond, 2018*). *Sirthenea flavipes* is found on the ground in grassland and farmland (*Ito, Okutani & Hiura, 1977*; *Tomokuni et al., 1993*; *Takahashi, 1996*; *Hirashima & Morimoto, 2008*). Similar to adults of the North American species *Sirthenea carinata* (Fabricius) (*Hudson, 1987*), juveniles and adults of *S. flavipes* exclusively prey on mole crickets (*Hayashi, 2023*); *S. flavipes* uses a long proboscis (labium) to inject paralyzing venoms into prey, then feeds on the prey (Video S2; Fig. 2). *Sirthenea flavipes* aggressively stabs attackers with its proboscis when it is caught (*Ito, Okutani & Hiura, 1977*; *Yasunaga et al., 2018*). A bite (stab) by adult *S. flavipes* reportedly causes severe pain to humans (*Takara, 1957*; *Ito, Okutani & Hiura, 1977*; *Tomokuni et al., 1993*; *Takahashi, 1996*; *Yasunaga et al., 2018*). The assassin bug *S. flavipes* shares the microhabitat (on the ground in grassland), prey (mole crickets), and body colour pattern (black and yellow colour) with the bombardier beetle *Ph. occipitalis jessoensis* (Fig. 1). However, the relationship between the bombardier beetle *Ph. occipitalis jessoensis* and assassin bug *S. flavipes* remains unexplored.

The black-spotted pond frog *Pe. nigromaculatus* is distributed in East Asia, including Japan, Korea, and China (*Komaki et al., 2015*; *Matsui & Maeda, 2018*). Postmetamorphic juveniles and adults of *Pe. nigromaculatus* are found in paddy fields and surrounding grasslands (*Matsui & Maeda, 2018*). Postmetamorphic juveniles and adults prey on animals smaller than themselves, especially terrestrial insects (*Hirai & Matsui, 1999*; *Hirai, 2002*; *Honma, Oku & Nishida, 2006*; *Sano & Shinohara, 2012*; *Sarashina, Yoshihisa & Yoshida, 2011*; *Sugiura, 2018*). *Pelophylax nigromaculatus* uses its tongue to catch and swallow prey

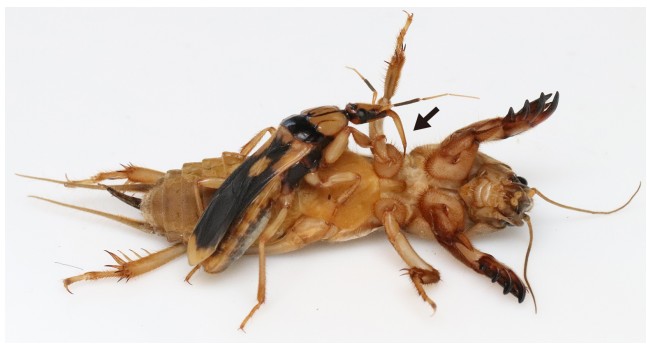

**Figure 2  An assassin bug *Sirthenea flavipes* feeding on the mole cricket *Gryllotalpa orientalis*.** The arrow indicates the proboscis of the adult assassin bug. Photo credit: Shinji Sugiura.

(*Sugiura, 2018*; *Sugiura, 2020b*; *Sugiura & Tsujii, 2022*). *Pelophylax nigromaculatus* is found with *Ph. occipitalis jessoensis* and *S. flavipes* on grasslands in central Japan (Fig. 1). *Sugiura (2018)* observed that almost all *Pe. nigromaculatus* juveniles and adults could attack the bombardier beetle *Ph. occipitalis jessoensis*, but ultimately rejected it under laboratory conditions. Therefore, the pond frog *Pe. nigromaculatus* was used as a model predator to investigate the mimetic interaction between *Ph. occipitalis jessoensis* and *S. flavipes*.

To elucidate the nature of the mimetic interaction between bombardier beetles and assassin bugs, we observed behavioural responses of the frog *Pe. nigromaculatus* to *Ph. occipitalis jessoensis* and *S. flavipes* under laboratory conditions. Specifically, we compared rates of rejection by *Pe. nigromaculatus* between *Ph. occipitalis jessoensis* and *S. flavipes*. To determine whether *Ph. occipitalis jessoensis* and/or *S. flavipes* benefits from the mimetic interaction, we experimentally investigated whether a frog would attack an insect species after it had encountered the other insect species. The rate of attack by frogs that had encountered bombardier beetles (or assassin bugs) was compared with the rate of attack by frogs that had not encountered the indicated species. Finally, we discuss the adaptive significance of mimetic interactions between bombardier beetles and assassin bugs.

## MATERIALS AND METHODS

### Sampling

Forty-three adults of the bombardier beetle *Ph. occipitalis jessoensis* were collected from Honshu (Hyogo, Shiga, and Shimane Prefectures) in July–September 2020 and May–October 2021. Adult beetles were housed separately in plastic cases (diameter: 85 mm; height: 25 mm) under laboratory conditions (25 °C; cf. *Sugiura, 2018*; *Sugiura & Sato, 2018*; *Sugiura, 2021*; *Sugiura & Date, 2022*) and fed dead larvae of *Spodoptera litura* (Fabricius) (Lepidoptera: Noctuidae) (cf. *Sugiura, 2018*; *Sugiura & Sato, 2018*; *Sugiura, 2021*; *Sugiura & Date, 2022*). Prior to experiments, body length and weight were measured to the nearest 0.01 mm and 0.1 mg using electronic slide callipers (CD-15AX, Mitsutoyo, Kawasaki, Japan) and an electronic balance (CPA64, Sartorius Japan K.K., Tokyo, Japan), respectively (Table 1). The same beetles were not used for multiple experiments.

**Table 1  Body sizes of bombardier beetles, assassin bugs, and pond frogs used in this study.**

|  | Bombardier beetle *Pheropsophus occipitalis jessoensis* | Assassin bug *Sirthenea flavipes* | Pond frog *Pelophylax nigromaculatus* |
|---|---|---|---|
| Body length (mm)[a] | 18.7 ± 0.2 (15.9–20.9) | 20.3 ± 0.2 (18.6–21.9) | 51.2 ± 0 .9 (39.0–62.0) |
| Body weight (mg)[a] | 291.9 ± 8.9 (160.2–412.9) | 133.9 ± 6.7 (92.9–201.7) | 13428.8 ± 828.0(5497.4–29761.3) |
| *n* | 43 | 20 | 48 |

Notes.
[a]Mean ± SE (range: minimum–maximum).

Twenty adults of the assassin bug *S. flavipes* were collected from Honshu (Hyogo and Shimane Prefectures) in August–October 2021. Adult bugs were housed separately in plastic cases (diameter: 85 mm; height: 25 mm) under laboratory conditions (25 °C). Prior to experiments, body length and weight were measured to the nearest 0.01 mm and 0.1 mg using electronic slide callipers and an electronic balance, respectively (Table 1). Adults and nymphs of the mole cricket *G. orientalis* were provided as prey (cf. *Hayashi, 2023*). Some assassin bugs were used repeatedly in different experiments.

Forty-eight adults and juveniles of the pond frog *Pe. nigromaculatus* were collected from Honshu (Hyogo Prefecture) in June–October 2021. Frogs were housed separately in plastic cages (length: 120 mm; width: 85 mm; height: 130 mm) in the laboratory at 25 °C (cf. *Sugiura, 2018*; *Sugiura, 2020b*; *Sugiura & Tsujii, 2022*). Live mealworms—larvae of *Tenebrio molitor* Linnaeus (Coleoptera: Tenebrionidae)—were provided as prey (cf. *Sugiura, 2018*; *Sugiura, 2020b*; *Sugiura & Tsujii, 2022*). Snout–vent length and body weight were measured to the nearest 0.01 mm and 0.1 mg using electronic slide callipers and an electronic balance, respectively (Table 1). Some frogs were used repeatedly in different experiments.

## Experiment 1: initial response

We used the predator *Pe. nigromaculatus* to test whether the bombardier beetle *Ph. occipitalis jessoensis* or the assassin bug *S. flavipes* is better defended under laboratory conditions. In accordance with the method established by *Sugiura (2018)*, we experimentally investigated behavioural responses of *Pe. nigromaculatus* to *Ph. occipitalis jessoensis* and *S. flavipes* in the laboratory (Graduate School of Agricultural Science, Kobe University) between September 2021 and October 2021. We used frogs that had fed on mealworms >24 h before experiments. First, a frog was placed in a plastic cage (length: 120 mm; width: 85 mm; height: 130 mm). Next, a bombardier beetle (or an assassin bug) was transferred to the cage containing the frog. The behaviours of the frog and the bombardier beetle (or assassin bug) were recorded using a digital camera (iPhone 12 Pro Max; Apple Inc., Cupertino, CA, USA) and a digital video camera (Handycam HDR-PJ790V, Sony, Tokyo, Japan). The footage of recorded behaviour was reviewed to investigate how each insect could defend. The bombing sounds of bombardier beetles were checked to investigate whether bombing forced the frogs to reject the beetles. Stabbing by assassin bugs was investigated to determine whether stabbing forced the frogs to reject the bugs. When a frog did not attack a bombardier beetle or an assassin bug within 2 min, we considered it to have ignored the insect. When a frog swallowed a bombardier beetle or an assassin bug, we observed

whether the frog subsequently vomited the insect (cf. *Sugiura & Sato, 2018*; *Sugiura, 2018*; *Sugiura & Date, 2022*). When the frog did not vomit the insect, we inferred that the frog had digested the insect. In Experiment 1, we used 20 bombardier beetles, 20 assassin bugs, and 40 frogs. The body size (snout–vent length and body weight) of frogs that attacked bombardier beetles did not significantly differ from the body size of frogs that attacked assassin bugs ($t$-test; snout–vent length, $t = -0.77455$, $df = 37.923$, $P = 0.4434$; body weight, $t = 0.98556$, $df = 35.179$, $P = 0.3311$). The same individuals of bombardier beetles, assassin bugs, and frogs were not used repeatedly in this experiment. The sample size was determined based on the number of assassin bugs collected in this study. Experiment 1 was part of the following experiment; specifically, the initial responses of 35 frogs observed in Experiment 2 were used as the data for Experiment 1.

**Experiment 2: generalisation tests**

We experimentally investigated the interaction between the bombardier beetle *Ph. occipitalis jessoensis* and the assassin bug *S. flavipes via* the potential predator *Pe. nigromaculatus* under laboratory conditions. Specifically, we investigated how a frog responded to a bombardier beetle or an assassin bug after the frog had encountered the other insect species (*i.e.*, generalisation test; Fig. 3). The same plastic cages and video cameras (see Experiment 1 for details) were used in this experiment. A bombardier beetle was provided to a frog that had encountered (attacked or ignored) an assassin bug ($n = 23$; Fig. 3A). We tested whether the frog attacked or ignored the bombardier beetle approximately 6 min (median: 6 min; range: 5–14 min) after the frog had encountered an assassin bug. The rate of attack on bombardier beetles by frogs that had encountered assassin bugs was compared to the rate of attack by frogs that had not encountered assassin bugs. Similarly, an assassin bug was provided to a frog that had encountered a bombardier beetle ($n = 20$; Fig. 3B). We tested whether the frog attacked or ignored the assassin bug approximately 6 min (median: 6 min; range: 5–7 min) after the frog had encountered a bombardier beetle. The rate of attack on assassin bugs by frogs that had encountered bombardier beetles was compared to the rate of attack by frogs that had not encountered bombardier beetles. Although the duration used in our generalisation tests (5–14 min) was shorter than the generalisation time of a spider (1 h; *Raška et al., 2020*) and the memory retention time of a bird (35 days; *Kojima & Yamamoto, 2020*), our field observations (Fig. 1) suggest that pond frogs frequently encounter bombardier beetles and assassin bugs under field conditions. Survivals of bombardier beetles, assassin bugs, and frogs used in this study were checked within 24 h after experiments. When a frog died within 24 h after the experiment, we dissected the frog to investigate the cause of death. In Experiment 2, we used 43 bombardier beetles, 17 assassin bugs, and 43 frogs. The same individuals of bombardier beetles and frogs were not used repeatedly in this experiment. The initial behavioural responses of 35 frogs observed in this experiment were also used as the data for Experiment 1. In addition, eight frogs were exclusively used in Experiment 2.

All experiments were performed in accordance with Kobe University Animal Experimentation Regulations (Kobe University's Animal Care and Use Committee, No. 30–01). Only one pair of insect species was provided to each frog to minimise the negative

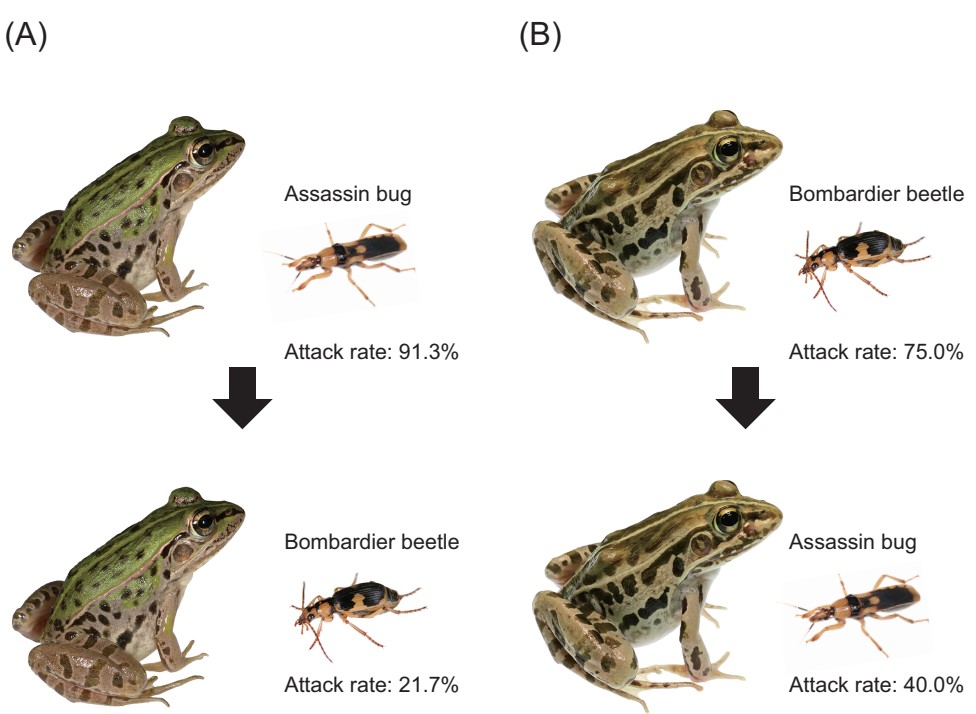

(A)   (B)

Assassin bug

Attack rate: 91.3%

Bombardier beetle

Attack rate: 75.0%

Bombardier beetle

Attack rate: 21.7%

Assassin bug

Attack rate: 40.0%

**Figure 3** **Experimental procedures and results of generalisation tests.** (A) A bombardier beetle was provided to a frog that had encountered an assassin bug. (B) An assassin bug was provided to a frog that had encountered a bombardier beetle. Photo credit: Shinji Sugiura.

impacts of well-defended insects on frogs. Healthy frogs were released after the experiments had been completed. No frogs were euthanised in this study.

## Data analysis

All analyses were performed using R version 3.5.2 (*R Core Team, 2018*).

Fisher's exact test was used to compare the rates of rejection by frogs between bombardier beetles and assassin bugs. Welch's $t$-test was used to compare the body size (body length and weight) of bombardier beetles and assassin bugs; it was also used to compare the body size (snout–vent length and body weight) of pond frogs that attacked bombardier beetles and assassin bugs. A generalised linear mixed model (GLMM) with a binomial error distribution and a logit link was used to investigate the effects of insect species and frog encounter history on the rate of attack by frogs. The frog response (attack, 1; or ignore, 0) was used as a response variable. The insect species (the bombardier beetle *Ph. occipitalis jessoensis* or the assassin bug *S. flavipes*), frog encounter history (an initial response or a response after encountering the other species), and the interaction between insect species and frog encounter history were used as fixed factors. Individual assassin bugs and frogs were used as random effects. The GLMM was conducted using the lme4 package version 1.1.19 in R (*Bates et al., 2015*). A significance threshold of 0.05 was used for all statistical tests.

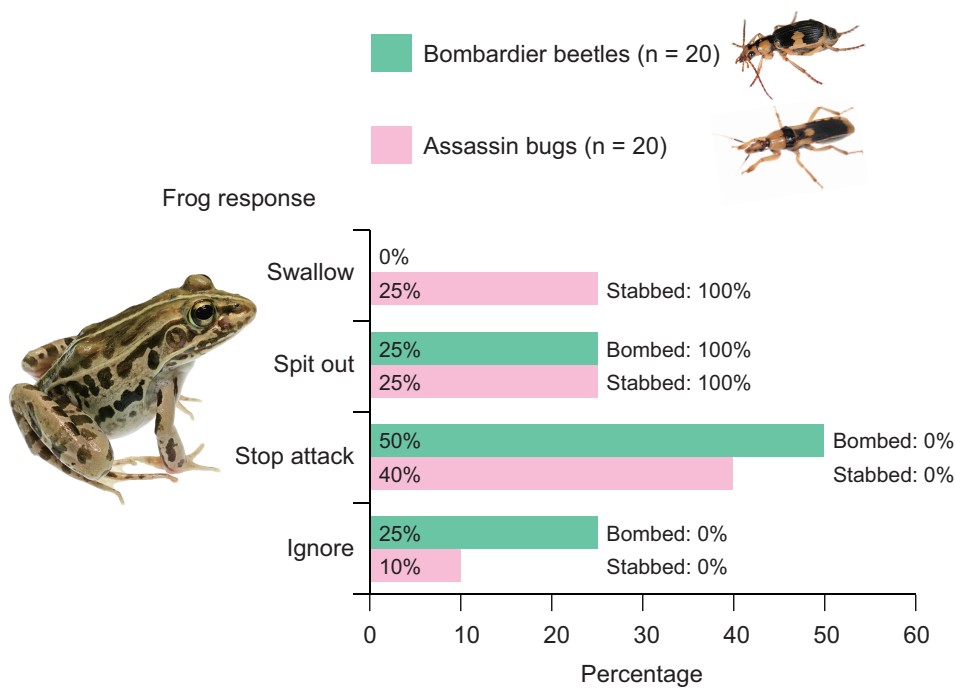

**Figure 4** **Behavioural responses of the frog *Pelophylax nigromaculatus* to the bombardier beetle *Pheropsophus occipitalis jessoensis* and the assassin bug *Sirthenea flavipes*.** Frogs that had not encountered the bombardier beetle or assassin bug were used in this study. Swallow: frogs successfully swallowed beetles (or bugs). Spit out: frogs spat out beetles (or bugs) immediately after taking the indicated insects into their mouths. Stop attack: frogs stopped their attacks immediately after their tongues had contacted beetles (or bugs). Ignore: frogs did not attack beetles (or bugs). Bombed: frogs were bombed by beetles. Stabbed: frogs were bitten (stabbed) by bugs. Photo credit: Shinji Sugiura.

## RESULTS

### Experiment 1: initial response

Among the 20 frogs in this experiment, none successfully swallowed bombardier beetles (Fig. 4). Five frogs (25%) captured beetles in their mouths but spat out the beetles immediately after bombing had occurred in their mouths (Video S3; Figs. 4 and 5). Ten frogs (50%) stopped attacking beetles immediately after their tongues had contacted the beetles (Fig. 4). Five frogs (25.0%) ignored the beetles (Fig. 4).

Five (25%) of 20 frogs successfully swallowed assassin bugs (Fig. 3), although these frogs were stabbed by assassin bugs in their mouths (Fig. 4). Fifteen of the 20 frogs (75%) rejected assassin bugs. Five frogs (25%) captured assassin bugs in their mouths but spat out the bugs immediately after stabbing had occurred in their mouths (Video S4; Figs. 4 and 6). Eight frogs (40%) stopped attacking assassin bugs immediately after their tongues had contacted the bugs (Fig. 4). Two frogs (10.0%) ignored the bugs (Fig. 4).

The rate of bombardier beetle rejection by frogs (100.0%) significantly differed from the rate of assassin bug rejection by frogs (75.0%; Fisher's exact test, $P = 0.0471$). The body size (body length and weight) significantly differed between bombardier beetles and assassin bugs; bombardier beetles were shorter but heavier than assassin bugs ($t$-test; body

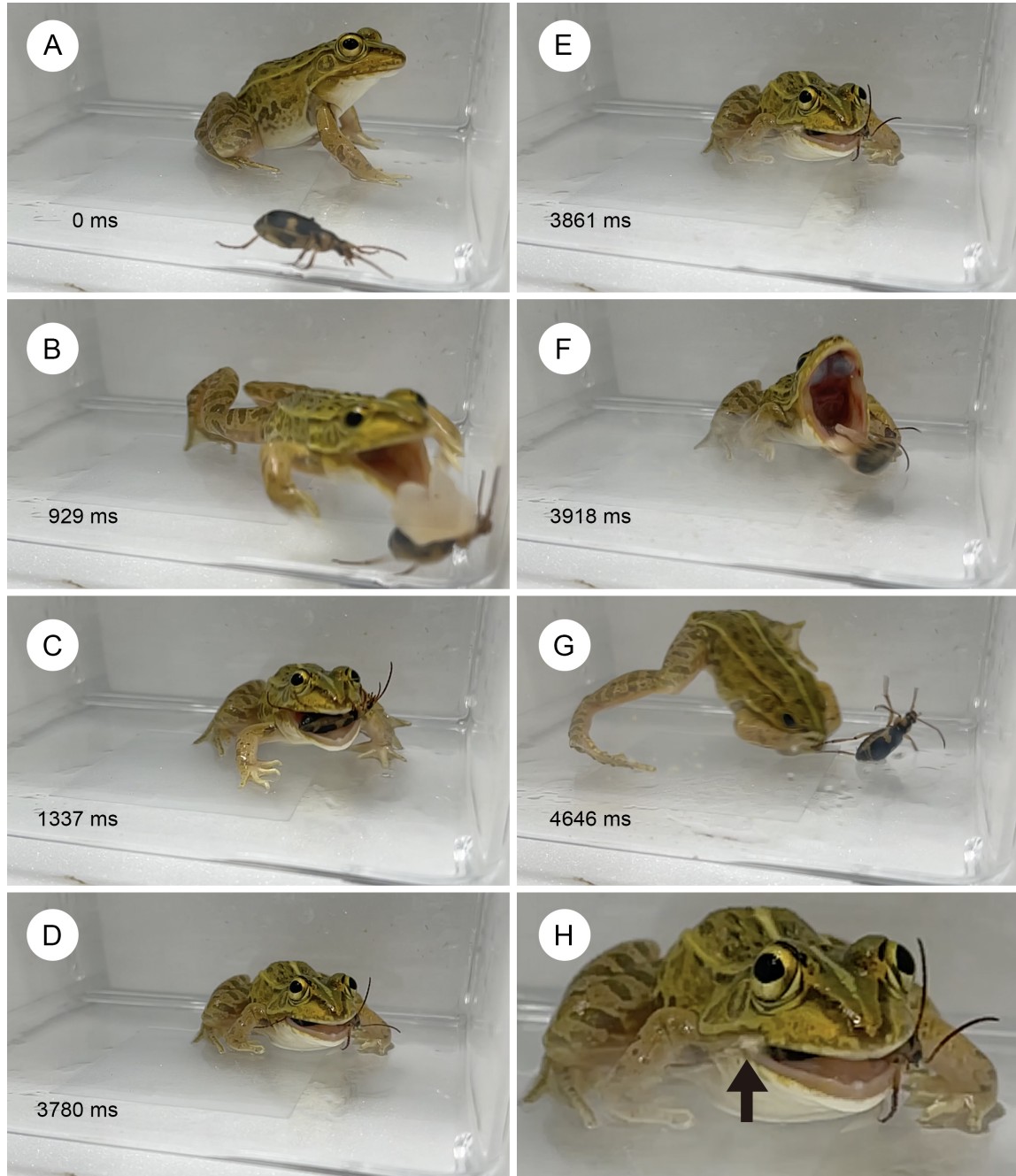

**Figure 5** Temporal sequence of the frog *Pelophylax nigromaculatus* rejecting an adult bombardier beetle *Pheropsophus occipitalis jessoensis.* (A) 0 ms. (B) 929 ms. (C) 1,337 ms. (D) 3,780 ms. (E) 3,861 ms. (F) 3,918 ms. (G) 4,646 ms. (H) Close-up view (E), with the arrow indicating bombing from the tip of the abdomen of the adult *Ph. occipitalis jessoensis.* The frog spat out the bombardier beetle after bombing had occurred in its mouth (see Video S3). Credit: Shinji Sugiura.

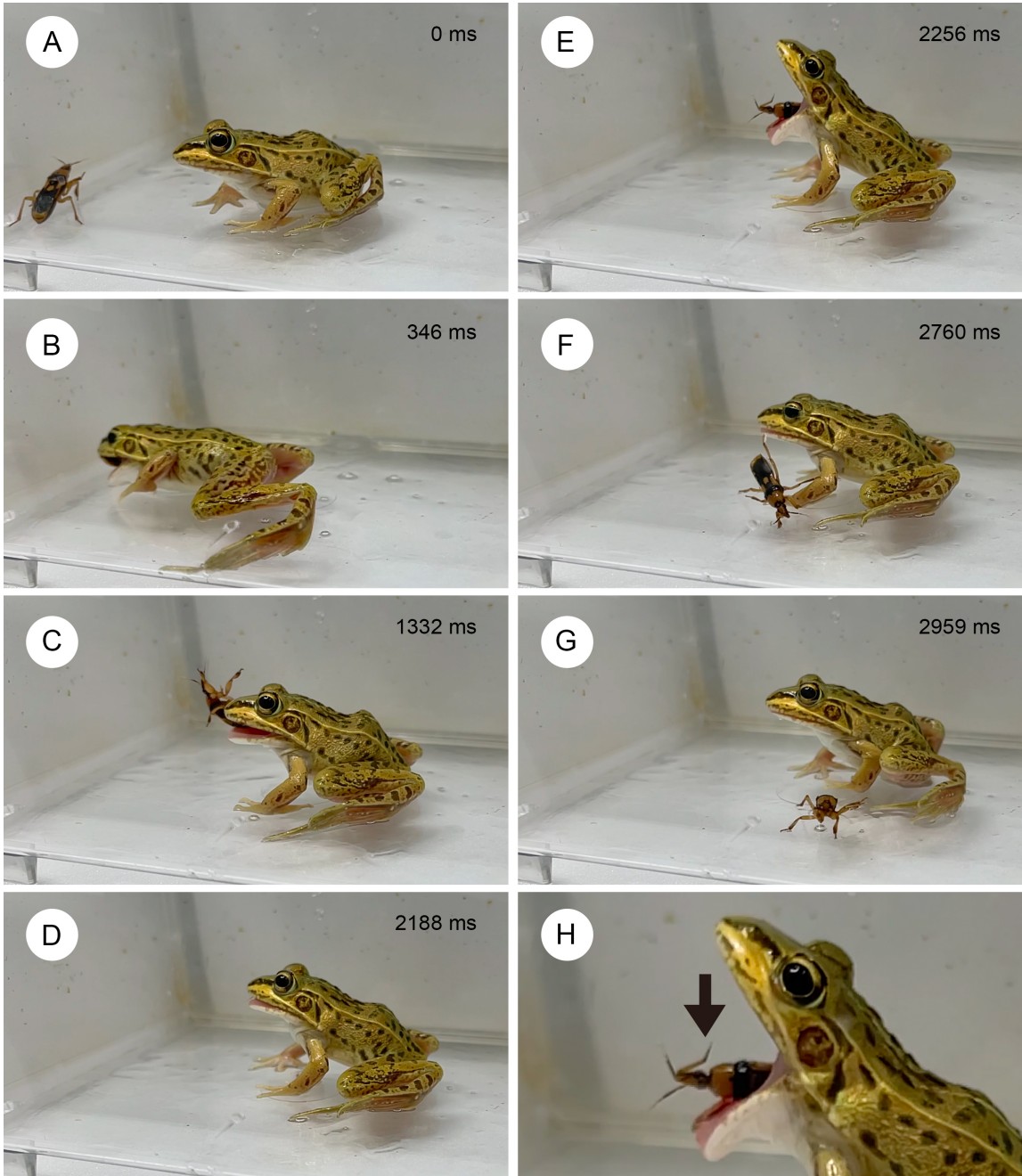

**Figure 6** **Temporal sequence of the frog *Pelophylax nigromaculatus* rejecting an adult assassin bug *Sirthenea flavipes*.** (A) 0 ms. (B) 346 ms. (C) 1,332 ms. (D) 2,188 ms. (E) 2,256 ms. (F) 2,760 ms. (G) 2,959 ms. (H) Close-up view (E), with the arrow indicating the proboscis of the adult assassin bug. The frog spat out the assassin bug after stabbing had occurred in its mouth (see Video S4). Credit: Shinji Sugiura.

**Table 2  Results of generalisation tests: responses of frogs to assassin bugs (first) and bombardier beetles (second).**

| | | Second trial: frog responses to bombardier beetles | | | | |
|---|---|---|---|---|---|---|
| | Frog behaviour[a] | Swallow | Spit out (bombed) | Stop attack | Ignore | Total |
| First trial: frog responses to assassin bugs | Swallow | 0 | 0 | 0 | 5 | 5 |
| | Spit out (stabbed) | 0 | 1 | 1 | 2 | 4 |
| | Stop attack | 0 | 0 | 3 | 9 | 12 |
| | Ignore | 0 | 0 | 0 | 2 | 2 |
| | Total | 0 | 1 | 4 | 18 | 23 |

Notes.

Values: numbers of frogs.

[a]Frog behaviour: Swallow: frogs successfully swallowed beetles (or bugs). Spit out: frogs spat out beetles (or bugs) immediately after capturing the insects in their mouths (frogs were bombed or stabbed). Stop attack: frogs stopped their attacks immediately after their tongues had contacted beetles (or bugs). Ignore: frogs did not attack beetles (or bugs).

**Table 3  Results of generalisation tests: responses of frogs to bombardier beetles (first) and assassin bugs (second).**

| | | Second trial: frog responses to assassin bugs | | | | |
|---|---|---|---|---|---|---|
| | Frog behaviour[a] | Swallow | Spit out (stabbed) | Stop attack | Ignore | Total |
| First trial: frog responses to bombardier beetles | Swallow | 0 | 0 | 0 | 0 | 0 |
| | Spit out (bombed) | 0 | 1 | 0 | 4 | 5 |
| | Stop attack | 0 | 1 | 5 | 4 | 10 |
| | Ignore | 0 | 0 | 1 | 4 | 5 |
| | Total | 0 | 2 | 6 | 12 | 20 |

Notes.

Values: numbers of frogs.

[a]Frog behaviour: Swallow: frogs successfully swallowed beetles (or bugs). Spit out: frogs spat out beetles (or bugs) immediately after capturing the insects in their mouths (frogs were bombed or stabbed). Stop attack: frogs stopped their attacks immediately after their tongues had contacted beetles (or bugs). Ignore: frogs did not attack beetles (or bugs).

length, $t = -4.8737$, $df = 35.532$, $P < 0.0001$; body weight, $t = 11.255$, $df = 30.211$, $P < 0.0001$).

## Experiment 2: generalisation tests

Twenty-one (91.3%) of 23 frogs attacked assassin bugs (Figs. 3A and 7; Table 2). Bombardier beetles were provided to the frogs that had encountered assassin bugs ($n = 23$); five frogs (21.7%) attacked bombardier beetles (Figs. 3A and 7; Table 2), while 18 frogs (78.3%) ignored bombardier beetles (Table 2).

Fifteen (75.0%) of 20 frogs attacked bombardier beetles (Figs. 3B and 7; Table 3). Assassin bugs were provided to the frogs that had encountered bombardier beetles ($n = 20$); eight frogs (40.0%) attacked assassin bugs (Figs. 3B and 7), while 12 frogs (60.0%) ignored assassin bugs (Table 3).

A history of assassin bug encounter reduced the rate of attack on bombardier beetles by frogs from 75.0% to 21.7% (Fig. 7), although the rate of rejection by frogs did not change (100%; Fig. 8A). A history of bombardier beetle encounter reduced the rate of attack on assassin bugs by frogs from 91.3% to 40.0% (Fig. 7); the rate of rejection increased from 78.3% to 100.0% (Fig. 8B). The GLMM analysis showed that frog encounter history had a significant influence on the rate of attack by frogs, although insect species and the

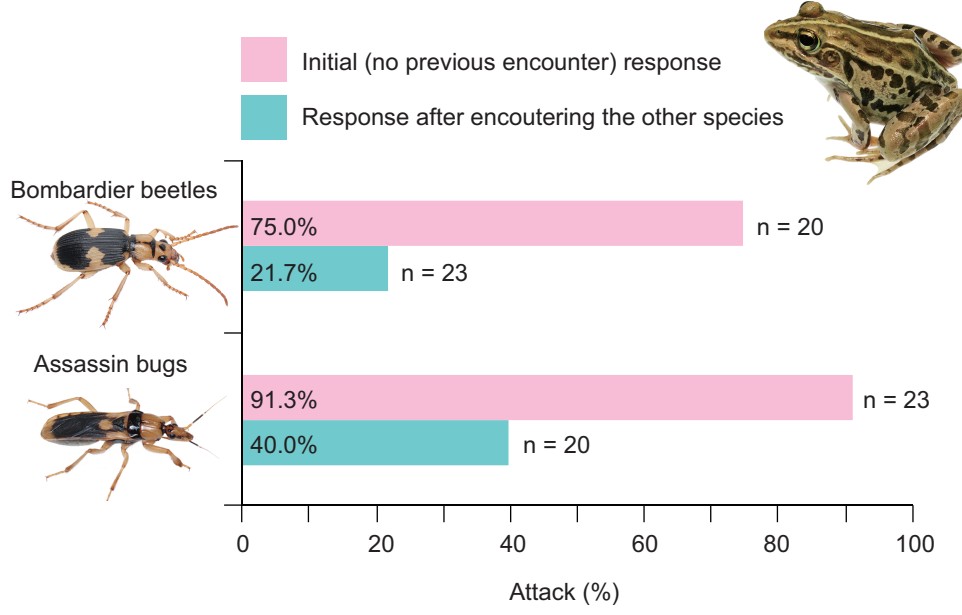

**Figure 7** Rates of attack on the bombardier beetle *Pheropsophus occipitalis jessoensis* and the assassin bug *Sirthenea flavipes* by the frog *Pelophylax nigromaculatus* before and after encounters with the other insect. Photo credit: Shinji Sugiura.

**Table 4** Results of a generalised linear mixed model (GLMM) to identify factors that influenced the rates of attack on bombardier beetles and assassin bugs by frogs.

| Response variable | Explanatory variable (fixed factor) | Coefficient estimate | SE | z value | P value |
|---|---|---|---|---|---|
| Attack | Intercept | 3.2264 | 1.4013 | 2.302 | 0.0213 |
| | Insect species (*vs.* bombardier beetles)[a] | −1.5950 | 1.2290 | −1.298 | 0.1943 |
| | Frog encounter history (*vs.* encounter)[b] | −3.8513 | 1.7386 | −2.215 | 0.0267 |
| | Insect species × frog experience | 0.2856 | 1.6187 | 0.176 | 0.8599 |

**Notes.**
[a] Assassin bugs were used as a reference.
[b] Initial (no previous encounter) response was used as a reference.

interaction between insect species and frog encounter history did not have significant effects on the rate of attack (Table 4).

## Survival

None of the bombardier beetles or assassin bugs that successfully defended against frogs died within 24 h after the experiments. Of the frogs ($n = 48$), one (2.1%) died within 24 h (4 h) after swallowing an assassin bug; a dead assassin bug was found in the stomach of the dead frog. Other frogs (97.9%) were not harmed by our experiments.

## DISCUSSION

There has been controversy regarding whether mimetic interactions between unequally defended species are parasitic (*Speed et al., 2000*; *Rowland et al., 2007*; *Rowland et al., 2010*;

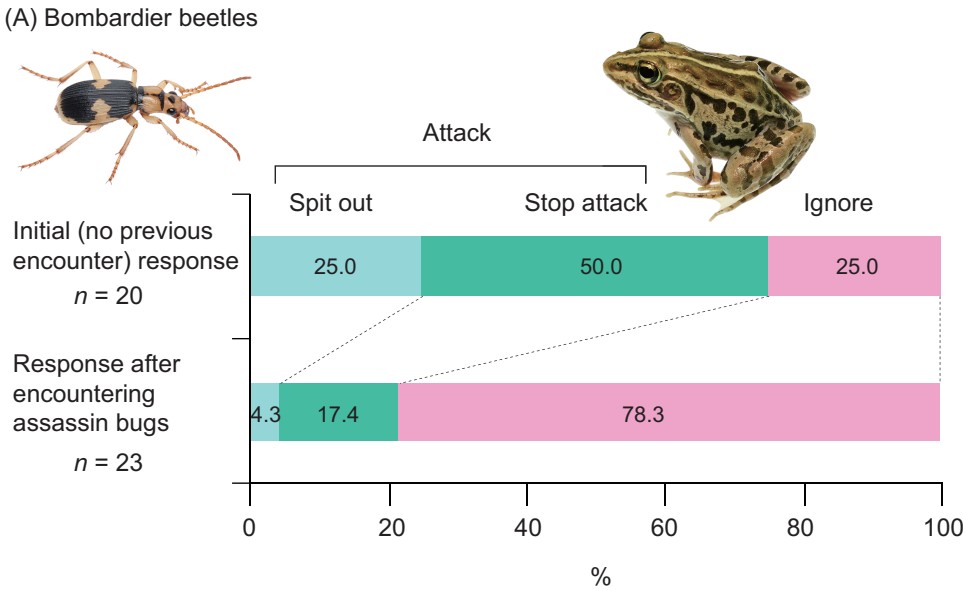

(A) Bombardier beetles

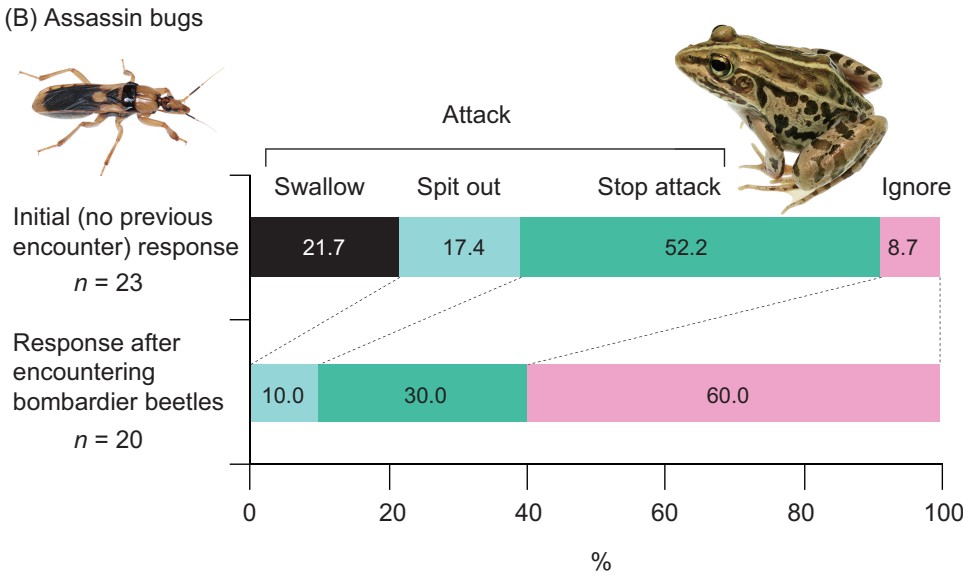

(B) Assassin bugs

**Figure 8** **Responses of the frog** *Pelophylax nigromaculatus* **to the bombardier beetle** *Pheropsophus oc-cipitalis jessoensis* **and the assassin bug** *Sirthenea flavipes* **after the frog encountered the other insect species.** Swallow: frogs successfully swallowed beetles (or bugs). Spit out: frogs spat out beetles (or bugs) immediately after capturing the insects in their mouths (frogs were bombed or stabbed). Stop attack: frogs stopped their attacks immediately after their tongues had contacted beetles (or bugs). Ignore: frogs did not attack beetles (or bugs). Photo credit: Shinji Sugiura.

*Aubier, Joron & Sherratt, 2017*). In the present study, we showed that both the bombardier beetle *Ph. occipitalis jessoensis* and the assassin bug *S. flavipes* were well-defended against the frog *Pe. nigromaculatus* (Fig. 4). In generalisation tests (Experiment 2), frogs with a history of assassin bug encounter attacked bombardier beetles less frequently compared with frogs

that had no such encounter history (Figs. 3A, 7 and 8A). Similarly, frogs with a history of bombardier beetle encounter attacked assassin bugs less frequently compared with frogs that had no such encounter history (Figs. 3B, 7 and 8B). These results suggest that both the bombardier beetle *Ph. occipitalis jessoensis* and the assassin bug *S. flavipes* benefit from the mimetic interaction in terms of defence against the potential predator *Pe. nigromaculatus*. Although *Ph. occipitalis jessoensis* demonstrated superior defensive abilities compared to *S. flavipes* (Figs. 4 and 8), the mimetic interaction between these two unequally defended species may be mutualistic, rather than parasitic.

## Frogs as predators

Frogs have been frequently used as predators to investigate the effectiveness of anti-predator defences in insects (*Taniguchi et al., 2005*; *Ito, Taniguchi & Billen, 2016*; *Matsubara & Sugiura, 2017*; *Sugiura, 2018*; *Shinohara & Takami, 2020*; *Sugiura, 2020a*; *Sugiura, 2020b*; *Sugiura & Date, 2022*; *Sugiura & Tsujii, 2022*). In this study, we used pond frogs as predators of bombardier beetles and assassin bugs to show that the frog species *Pe. nigromaculatus* was unable to distinguish between the bombardier beetle *Ph. occipitalis jessoensis* and the assassin bug *S. flavipes*. This finding could be explained by the adaptive generalisation in predators, where they learn to recognise dangerous prey by generalising the appearance of previously encountered prey to that of subsequently encountered prey (*Ruxton et al., 2008*).

In the Experiment 2, the time used in the generalisation tests ranged from 5–14 min. The time used in previous memory and generalisation tests (1 h–35 days; *Ito, Taniguchi & Billen, 2016*; *Kojima & Yamamoto, 2020*; *Raška et al., 2020*) was longer than the time of our generalisation tests. For example, the tree frog *Dryophytes japonica* (Günther) reportedly retains the memory of unpalatable prey for at least 1 day (*Ito, Taniguchi & Billen, 2016*). In addition, the *Pe. nigromaculatus* individuals used in this study were collected from the study sites where both *Ph. occipitalis jessoensis* and *S. flavipes* were found, suggesting that some individuals of *Pe. nigromaculatus* may have already experienced *Ph. occipitalis jessoensis* and/or *S. flavipes* at the sites prior to our experiments. Therefore, the use of short experimental durations and wild-collected individuals may have influenced the results of our experiments in *Pe. nigromaculatus*. Further studies are needed to investigate detailed memory retention in *Pe. nigromaculatus*.

Some predators have evolved counter defences, such as specific skills to avoid well-defended prey by detecting toxic chemicals or recognising warning signals (*Edmunds, 1974*; *Endler, 1991*; *Ruxton, Sherratt & Speed, 2004*; *Skelhorn & Rowe, 2006*; *Williams et al., 2010*). In the present study, 50% and 40% of frogs stopped attacking *Ph. occipitalis jessoensis* and *S. flavipes* before they had been bombed and stabbed, respectively (Fig. 4). Because *Pe. nigromaculatus* individuals stopped attacking immediately after their tongues had contacted these insects, this frog species may quickly detect deterrent characteristics on the body surfaces of *Ph. occipitalis jessoensis* and *S. flavipes* with its tongue; this enables avoidance of damage (*Sugiura, 2018*). Such reactions to well-defended prey have been reported in other predators such as tree frogs (*Ito, Taniguchi & Billen, 2016*) and quails (*Kojima & Yamamoto, 2020*).

## Bombardier beetles as models and mimics

Bombardier beetles can chemically defend themselves against various types of predators (*Eisner, 1958*; *Eisner & Meinwald, 1966*; *Eisner & Dean, 1976*; *Dean, 1980*; *Eisner et al., 2006*; *Bonacci et al., 2008*; *Sugiura & Sato, 2018*; *Sugiura, 2018*; *Kojima & Yamamoto, 2020*; *Sugiura, 2021*; *Sugiura & Date, 2022*). Many bombardier beetle species have aposematic body colour patterns that advertise their toxicity to predators (*Schaller et al., 2018*; *Anichtchenko et al., 2022*). Therefore, bombardier beetles are visually mimicked by distantly related insects that coexist with them in the same habitats (*Shelford, 1902*; *Bonacci et al., 2008*; *Kojima & Yamamoto, 2020*). However, very few studies have elucidated the nature of mimetic interactions that include bombardier beetles. In the present study, we used the frog *Pe. nigromaculatus* as a potential predator to investigate that the mimetic interaction between the bombardier beetle *Ph. occipitalis jessoensis* and the assassin bug *S. flavipes*. We found that a history of encounter with *Ph. occipitalis jessoensis* reduced the rate of attack on *S. flavipes* (Fig. 8B), suggesting that the coexistence with *Ph. occipitalis jessoensis* is beneficial for *S. flavipes*. However, *Ph. occipitalis jessoensis* consistently repelled the frog *Pe. nigromaculatus* in our study (Figs. 4 and 8A). Therefore, a history of encounters with the assassin bug *S. flavipes* may not benefit *Ph. occipitalis jessoensis*. Nevertheless, the mortality risk from frog attacks is not zero, as a previous study reported that 3.6% of the frog *Pe. nigromaculatus* successfully ate *Ph. occipitalis jessoensis* (*Sugiura, 2018*). Thus, the coexistence with the assassin bug *S. flavipes* is beneficial for the bombardier beetle *Ph. occipitalis jessoensis*, although the mutualistic interaction between these two insect species may be asymmetric.

## Assassin bugs as mimics

Ground-dwelling assassin bugs that belong to the subfamily Peiratinae reportedly stab with their proboscises, causing severe pain in humans (*Readio, 1927*; *Ito, Okutani & Hiura, 1977*; *Willemse, 1985*; *Tomokuni et al., 1993*; *Gil-Santana, Forero & Weirauch, 2015*; *Yasunaga et al., 2018*). Assassin bugs can paralyze prey and repel enemies through the injection of saliva or venom (*Eisner, Eisner & Siegler, 2005*; *Schmidt, 2009*; *Walker et al., 2016*). Assassin bugs also have scent glands to chemically defend themselves against predators (*Louis, 1974*; *Staddon, 1979*). However, few studies have investigated the effectiveness of anti-predator defences in assassin bugs (*Walker et al., 2018*; *Walker et al., 2019*).

In this study, we showed that the frog species *Pe. nigromaculatus* frequently rejected the assassin bug *S. flavipes* (Figs. 4 and 8). Although some frogs successfully swallowed *S. flavipes* individuals (Figs. 4 and 8B), one frog died 4 h after a successful swallowing event. These results suggest that *S. flavipes* venom is sufficiently strong to repel predators. In addition, some frogs stopped attacking assassin bugs immediately after their tongues had contacted the bugs (Fig. 4), suggesting that chemicals on the body surfaces of *S. flavipes* act as a deterrent to *Pe. nigromaculatus*.

Some assassin bug species share body colour patterns with hymenopteran insects such as paper wasps, ichneumonid wasps, spider wasps, and stingless bees (*Maldonado Capriles & Lozada Robles, 1992*; *Zhang & Weirauch, 2014*), suggesting that they mimic wasps (*Haviland, 1931*; *Forero & Giraldo-Echeverry, 2015*; *Gil-Santana, Forero & Weirauch,*

*2015*) and bees (*Jackson, 1973*; *Wattanachaiyingcharoen & Jongjitvimol, 2007*; *Gil-Santana, 2008*; *Gil-Santana, Forero & Weirauch, 2015*; *Alvarez, Zamudio & Melo, 2019*). Although assassin bugs reportedly coexist with model wasps or bees in the same microhabitats (*Alvarez, Zamudio & Melo, 2019*), the mimetic interactions between assassin bugs and other insects have not been experimentally tested using predators. In this study, we used the frog *Pe. nigromaculatus* as a potential predator to investigate the mimetic interaction between the assassin bug *S. flavipes* and the bombardier beetle *Ph. occipitalis jessoensis*. Although both species were well defended against predators, *S. flavipes* showed poorer defence than did *Ph. occipitalis jessoensis* (Figs. 4 and 8). This could be explained by the differences in body size between the two insect species (Table 1), as prey weight could influence predation success by the frog *Pe. nigromaculatus* (*Sugiura, 2018*).

The distribution of the assassin bug *S. flavipes* overlaps with the distribution of the bombardier beetle *Ph. occipitalis jessoensis* in East and Southeast Asia (*Chłond, 2018*; *Fedorenko, 2021*). However, *S. flavipes* is also found in South and West Asia where *Ph. occipitalis jessoensis* is not distributed (*Chłond, 2018*; *Fedorenko, 2021*). In the assassin bug *S. flavipes*, the body colour pattern of the South and West Asian populations partially differs from the body colour pattern of the East and Southeast Asian populations; the pronotum of the South and West Asian populations is redder than the pronotum of the East and Southeast Asian populations, although both types share the black and yellow pattern on other body parts (*Chond, Bugaj-Nawrocka & Sawka-Gadek, 2019*). Notably, the body colour pattern of South and West Asian *S. flavipes* is very similar to the body colour pattern of another bombardier beetle, *Pheropsophus* (*Stenaptinus*) *catoirei* (Dejean); adult *Ph. catoirei* individuals have a reddish head and pronotum (*Fedorenko, 2021*). *Pheropsophus catoirei*, which is closely related to *Ph. occipitalis jessoensis*, shares its distribution area (South and West Asia) with *S. flavipes* in East and Southeast Asia (*Chond, Bugaj-Nawrocka & Sawka-Gadek, 2019*; *Fedorenko, 2021*). Therefore, the mimetic partner of *S. flavipes* could differ between East–Southeast Asia and West–South Asia.

## CONCLUSIONS

Some aposematic species form 'mimicry rings' (*Kunte, Kizhakke & Nawge, 2021*; *Chatelain et al., 2023*). Mimicry rings are composed of at least two Müllerian co-mimics or one aposematic species plus one Batesian mimic (*Kunte, Kizhakke & Nawge, 2021*); the smallest mimicry rings include only two species (*Kunte, Kizhakke & Nawge, 2021*), while the largest mimicry rings include >100 species (*Pekár et al., 2017*). Although many studies have investigated mimicry rings that are composed of closely related taxa (*Kunte, Kizhakke & Nawge, 2021*), fewer studies have focused on mimicry complexes that involve distantly related taxa (*Linsley, Esiner & Klots, 1961*; *Pekár et al., 2017*). Our results suggest that the bombardier beetle *Ph. occipitalis jessoensis* and the distantly related *S. flavipes* form a multi-order mimetic complex. Other insects such as the rove beetle species *Ocypus weisei* Harold (Coleoptera: Staphylinidae) may be included in the mimetic complex; *O. weisei* adults share a similar microhabitat (on the ground in grassland), body colour pattern (black and yellow pattern), and movement pattern with *Ph. occipitalis jessoensis* and *S.*

*flavipes* adults in Japan. Furthermore, the black and yellow body colour pattern of these insects is similar to the typical aposematic colour of stinging hymenopteran insects, such as paper wasps and bees (*Chatelain et al., 2023*). The presence of a colour pattern similar to a typical aposematic pattern likely provides *S. flavipes* and *Ph. occipitalis jessoensis* with more robust protection from predators, compared with other colour patterns.

## ACKNOWLEDGEMENTS

We thank T Date for assisting with insect and frog maintenance. We also thank J Raška and anonymous reviewers for helpful comments on an earlier version of the manuscript.

### Funding

This study was supported by a Grant-in-Aid for Scientific Research (JSPS KAKENHI Grant number 19K06073) and funds from the Hoshizaki Green Foundation. The funders had no role in study design, data collection and analysis, decision to publish, or preparation of the manuscript.

### Grant Disclosures

The following grant information was disclosed by the authors:
Grant-in-Aid for Scientific Research: 19K06073.
Hoshizaki Green Foundation.

### Competing Interests

The authors declare there are no competing interests.

### Author Contributions

- Shinji Sugiura conceived and designed the experiments, performed the experiments, analyzed the data, prepared figures and/or tables, authored or reviewed drafts of the article, and approved the final draft.
- Masakazu Hayashi conceived and designed the experiments, prepared figures and/or tables, and approved the final draft.

### Animal Ethics

The following information was supplied relating to ethical approvals (*i.e.*, approving body and any reference numbers):

The experiments were performed in accordance with the Kobe University Animal Experimentation Regulations (Kobe University's Animal Care and Use Committee, No. 30–01).

### Field Study Permissions

The following information was supplied relating to field study approvals (*i.e.*, approving body and any reference numbers):

Our study was not conducted in any protected areas. Because the study insect and frog species were not protected, no specific permissions were required to collect these species in Japan.

## Data Availability

The raw data are available at figshare: Sugiura, Shinji; Hayashi, Masakazu (2023): Data from: Bombardiers and assassins: mimetic interactions between unequally defended insects. figshare. Dataset. https://doi.org/10.6084/m9.figshare.19636617.

## Supplemental Information

Supplemental information for this article can be found online at http://dx.doi.org/10.7717/peerj.15380#supplemental-information.

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
