# Peer review of "Bombardiers and assassins: mimetic interactions between unequally defended insects"

_PeerJ, doi:10.7717/peerj.15380_

## Round 0.1 · original submission · Major Revisions

Dear Drs. Sugiura and Hayashi:

Thanks for submitting your manuscript to PeerJ. I have now received three independent reviews of your work, and as you will see, the reviewers raised some concerns about the research. Despite this, these reviewers are optimistic about your work and the potential impact it will have on research studying insect ecology and behavior. Thus, I encourage you to revise your manuscript, accordingly, taking into account all of the concerns raised by both reviewers.

While the concerns of the reviewers are relatively minor, this is a major revision to ensure that the original reviewers have a chance to evaluate your responses to their concerns. There are many suggestions, which I am sure will greatly improve your manuscript once addressed.

I look forward to seeing your revision, and thanks again for submitting your work to PeerJ.

Good luck with your revision,

-joe

Reviewer 1 ·

Basic reporting

This paper demonstrated that P. occipitalis and S. flavipes were unequally chemically defended. The authors also showed that the mimetic interactions were mutualistic by memory tests. In general I like the topic of manuscript in focus and enjoyed reading it and watching movies. The text has a nice structure and the storyline, including the objectives and hypotheses, are easy to follow from my viewpoint. I have a few remarks that I would like to share with the authors for consideration.

1. As the authors note, the memory time in Experiment 2 is very short compared to other related studies. How did the authors determine the memory time? Please explain. I guess it is determined based on preliminary experiments. Do the frogs lose memory of unpalatable prey after more than 30 min or more?

2. The authors used field-collected frogs as a predator. If the frogs memorize the distasteful prey for long time (e.g., one week), this would be problematic for using the wild, non-naive animals in memory tests. The authors need to rationalize it.

3. The authors discuss the cognitive limitations of frogs in Discussion. Alternative explanation is adaptive generalization of predators (see Ruxton et al. 2008, doi: 10.1111/j.1558-5646.2008.00485.x), although it is difficult to test (this is “psychology” of predators).

Experimental design

no comment

Validity of the findings

no comment

·

Basic reporting

No comment.

Experimental design

No comment.

Validity of the findings

No comment.

Additional comments

The article concerns the highly relevant topic of mimetic relationships between unequally defended prey. As the authors thoroughly discuss in the introduction, this topic is still problematic despite numerous previous studies focused on it. Most studies of unequally defended prey use artificially defended prey, so the use of natural prey in the presented article fills a gap left by previous studies.
I appreciate the clarity of the experimental design, which perfectly fits the tested hypotheses. The prey species are well selected thanks to the expertise of the corresponding author. Overall, the article is well written and structured and has definitely improved my understanding of mimetic relationships of natural prey. I have only a very few concerns which I consider critical:

201 - What is called a "memory test" in the manuscript is actually a typical generalization test (e.g. Gamberale & Tullberg 1996, Svádová et al. 2009, Raška et al. 2020). Unfortunately, that leads to incorrect parallels to other studies (lines 216-220: Kojima & Yamamoto 2020: no generalization test took place; Raška et al. 2020: generalization tests were performed after 1 hour, not 1 day).

226 - I am sorry, but I am a bit confused: if 40 frogs were used in Experiment 1 (line 196), how could you use 43 frogs in experiment 2, especially when "experiment 2 included experiment 1" (line 227)? Fig. 7 does not shed light on the problem.

313-329 - This part seems particularly problematic to me. It discucces topics not presented in the introduction, the insight into Pelophylax vision is superficial, the interpretation of reactions to the control prey is rather vague etc. I would suggest rewriting this part so it would chase fewer rabbits, so to speak, for the sake of consistency of the manuscript.


Other than that, I have only minor comments:

88 - I understand that the "co-mimics" is an elegant common term when talking about both prey species, but I think it should be mentioned somewhere that the co-mimetic relationship was only hypothetical (or potential) before you performed the experiments. It was also possible that the relationship would be Batesian, in wich case it would be a model-mimic relationship.

125-134 - Could it be that aside from defending themselves by proboscis, the assassin bugs possess some chemical defence found in many other true bug taxa? That may explain results in lines 260-262 (frogs touched the bugs and immediately rejected them).

138-139 - If I may suggest slight reformulation: "smaller than themselves, especially terrestrial insects".

220-222 - What was the aim of presentation of the control prey? The results are mentioned in lines 277-278 and 282, but are not statistically analyzed and do not have any effect on other parts of the experiments (such as exclusion of data obtained from predators that did not consume the control prey, Raška et al. 2020). Discussion of these results may also be problematic (lines 321-325, see below). I think these results could be even excluded from the paper.

254-255 - Do you have any idea why the frogs rejected the beetles even though the beetles did not use their primary means of defence?

266-267 - Did prey size affect the frogs' predatory behaviour? I think the reader might be interested in that information rather than whether the prey species were of the same size or not.

267 - The P-values seem to be a bit too detailed, especially the latter one. Three or four decimals are usually sufficient (with lower P-values expressed as P < 0.001 or 0.0001).

267-271 - These results would better suit Methods chapter, as they test only the distribution of predators between groups, not any hypothesis linked to the main topic of the article.

277-278 - It might be useful to know whether the frogs that ignored the control prey responded to the experimental prey. Could it be that some/all of them ignored the presented prey due to their physiological condition (especially satiation level)?

288-289 - Your results have shown no asymmetry in generalization between the prey species although their defences were not equal. I think that might deserve a brief mention in the discussion.

346-353 - I would be careful when discussing mimicry in very closely related species. According to Zrzavý (1994), synapomorphic warning signalization (such as that seen in Brachinus spp.) has completely different evolutionary dynamics than signalization used in Batesian/Müllerian mimicry.

412 - Could "movement pattern" be a more suitable term than "walking behaviour"?

Fig. 2 - Although the photograph is impressive, it is not linked to the main topic of the article. The figures look great overall, I especially appreciate the illustrative value of Figs. 5 and 6.

Reviewer 3 ·

Basic reporting

Yes, clear and unambiguous, literature well organised and formatted. Good article structure except that lines 116-151 belong in Intro rather than Materials and Methods. Relevant results to hypotheses.

Experimental design

Original primary research. Question well-defined, relevant. Rigorous investigation, methods described in sufficient detail.

Validity of the findings

Valid and meaningful findings. Underlying data provided and analysis sound.

Additional comments

Review of PeerJ 73218-v0

This study looks at a mimicry ring between an assassin bug and a bombardier beetle, aiming to determine if it is mutualistic or parasitic. The authors use controlled predation studies in the lab to show that under their conditions, it appears to be mutualistic. It is a very nice piece of work and on the whole has been carefully executed and presented. I very much enjoyed the photos and especially the video of the frog rejecting the bombardier. I suggest it is suitable for publication after addressing the following minor items:

• Line 54: 'interactions they are' should be 'interactions are'
• Lines 116–151: This is nice background, but it would fit better in the Introduction than under the Materials and Methods
• Line 317: Surely 'in low light conditions' would be better than 'in the dark'
• Line 373: What would be the value of mimicking a stingless bee? Seems an odd one to put in this list
• Figure 4 legend: Since wild-caught frogs were used, how are the authors sure that 'Frogs that had not encountered the bombardier beetle or assassin bug were used in this study'. If this is right this should be discussed in the Discussion.

---

## Round 0.2 · accepted · Accept

Dear Drs. Sugiura and Hayashi:

Thanks for revising your manuscript based on the concerns raised by the reviewers. I now believe that your manuscript is suitable for publication. Congratulations! I look forward to seeing this work in print, and I anticipate it being an important resource for groups studying insect ecology and behavior. Thanks again for choosing PeerJ to publish such important work.

Best,

-joe